# Impact of bariatric surgery on premenopausal women's womanliness: A qualitative systematic review and meta-synthesis

Rebecca Paul [1,2,3], Ellen Andersson[1‡], Torsten Olbers[1,4‡], Jessica Frisk[1‡], Carina Berterö[5]

**1** Department of Surgery and Department of Biomedical and Clinical Sciences, Linköping University, Norrköping, Sweden, **2** Department of Surgery, Falun County Hospital, Falun, Sweden, **3** Centre for Clinical Research, Uppsala University, Region Dalarna, Falun, Sweden, **4** Wallenberg Centre for Molecular Medicine, Department of Biomedical and Clinical Sciences, Linköping University, Linköping, Sweden, **5** Division of Nursing Sciences and Reproductive Health, Department of Health, Medicine and Caring Sciences, Linköping University, Linköping, Sweden

☯ These authors contributed equally to this work.
‡ EA, TO and JF also contributed equally to this work.
* rebecca.paul@liu.se

**Data Availability Statement:** All relevant data are within the manuscript and its Supporting Information files.

## Abstract

### Introduction

Obesity is associated with several co-morbidities in women, including disturbed sex hormone regulation with menstrual disturbances, subfertility, hirsutism, and central fat dispersion, all with an impact on sexual function and quality of life. There are few investigations regarding women's experiences of obesity-related altered sex hormone regulation and resolution after bariatric surgery.

### Objectives

This systematic review and interpretive meta-synthesis aim to identify the current qualitative knowledge base concerning women undergoing bariatric surgery and experiences of changes after weight loss, emphasising aspects of womanliness.

### Methods

A systematic review and qualitative meta-synthesis was conducted to gain a deeper and broader understanding of the available knowledge about premenopausal women's experienced changes after bariatric surgery. Relevant papers were identified by systematically searching PubMed, CINAHL, Embase, PsycInfo, PsycArticles, Scopus, Cochrane Library, Web of Science and Open Grey. The quality of the included studies was assessed, and the data was interpreted and synthesised using Gadamer's hermeneutics. The review protocol was registered on PROSPERO (CRD42023394225).

### Results

A total of 10 studies were considered relevant and included in the qualitative meta-synthesis. Three fusions were identified and interpreted as: "Womanliness," "A healthy and

**Funding:** RP received funding from the County Council of Dalarna, Sweden, through research grant number 974932. The funder website: https://www.regiondalarna.se/plus/forskning/centrum-for-klinisk-forskning-dalarna—ckf/ The funders had no role in study design, data collection and analysis, decision to publish, or preparation of the manuscript. TO, EA, JF and CB received no specific funding for this work.

**Competing interests:** I have read the journal's policy and the authors of this manuscript have the following competing interests: TO participated in advisory boards and educational activities for Johnson & Johnson and Novo Nordisk unrelated to the submitted article, and reimbursements were directed to his academic institution. RP, EA, JF and CB declare that no competing interests exist. This does not alter our adherence to PLOS ONE policies on sharing data and materials.

functioning body," and "Mind and Body Connection." Women experienced a return to womanliness after undergoing bariatric surgery with restored menstruation cycles, improved fertility and changed hair and fat dispersion signalling restored sex hormones. Women value a return to a healthy and functioning body that improves their experience of life and ability to take part in it. However, women experienced difficulties in adapting mentally to the drastic physical changes that occur after undergoing surgery.

## Conclusions

Women that have undergone bariatric surgery report several benefits to their health and well-being, although difficulties in adapting mentally to changes in outer appearance need to be managed in order to successfully move forward with a new life after surgery.

## Introduction

The global prevalence of overweight and obesity is increasing and is currently considered a global epidemic with consequences on health and longevity [1]. Among the known co-morbidities of obesity are cardiovascular disease [2, 3], type 2 diabetes [4], osteoarthritis [5], sleep apnea [6, 7] and several forms of cancer [8, 9]. Living with the effects of obesity on health may potentially diminish quality of life and well-being [10, 11]. Endocrine and metabolic alterations resulting from obesity also influence sex hormone regulation and balance [12].

Altered sex hormone levels in women may substantially impact function and lead to complications such as a disrupted menstrual cycle, subfertility and difficulties involving conception and birth. In addition to effects on fertility, women with disrupted sex hormones also suffer from physical signs including hirsutism, acne and central fat distribution leading to male pattern hair and body dispersion [13]. The accumulation of these health effects can substantially impact a women's health and well-being, encompassing feelings of womanliness, sexual function, quality of life, and self-perception [14, 15].

Bariatric surgery, for example Roux-en-Y gastric bypass surgery, is an effective obesity treatment [16, 17]. Previous studies have demonstrated a reduction in co-morbidities of obesity as well as a substantial improvement in physical function and health-related quality of life following weight loss after bariatric surgery [18]. Many of the hormonal disturbances and physical signs of hyperandrogenism are resolved secondary to normalised sex hormone levels with improvement in fertility and sexual function [19, 20].

Qualitative research provides an in-depth and rich exploration of the experiences of women living with obesity and the consequences of altered sex hormone levels. Using qualitative research method, we move beyond the limitations that exist in quantitative techniques by not being limited to explicit, singular results without detail or elaboration. The interview situation allows participants to share their stories and describe their lived experiences in detail, including how these experiences have meaning to them in their context. Currently, there are few studies focused on the lived experiences of women who have undergone bariatric surgery and experienced changes in their sex hormones. Conducting a systematic review and meta-synthesis of qualitative studies would provide a more rigorous understanding of the current data. This study involves a systematic review and interpretive meta-synthesis to examine the situation for women who have undergone bariatric surgery and its effects on their womanliness including sex hormones, quality of life, sexual function, and self-perception. The objective is to gain a better understanding of the topic through a thorough analysis of the current literature.

## Materials and methods

### Study approach

The study was registered in 2023 in the International Prospective Register of Systematic Reviews (PROSPERO) database under registration number CRD42023394225 [21]. The focus of the analysis is an identification and integration of information about effects of bariatric surgery in premenopausal women rather than meta-analysis or meta-synthesis of the findings of studies [22]. The approach used is a qualitative meta-synthesis, i.e., an interpretive integration of qualitative findings that are themselves an interpretive synthesis of data comprising coherent and integrated descriptions or explanations of events, concepts, or phenomena [23]. A meta-synthesis is an integration that offers more than the sum of the individual data sets because it provides an innovative interpretation of the findings. To ensure accuracy, the PRISMA (Preferred Reporting Items for Systematic Reviews and Meta-analyses) guidance was followed throughout this review [24]. Please see S1 Table for the PRISMA checklist.

### Criteria for inclusion

The review aimed to explore the experiences reported by premenopausal women who have undergone bariatric surgery for obesity, primarily Roux-en-Y gastric bypass, although all forms of bariatric surgery were included. The review did not have any restrictions on the setting and included sources from 2000–2023. The primary focus was on women´s experiences and perceptions concerning changes in sex hormones after bariatric surgery such as changes in body shape, menstruation, hirsutism, vasomotor symptoms. Specifically, the review explored women's reflections on fertility and sexuality after undergoing bariatric surgery. Additionally, the review considered other perceived experiences of changes in life in relation to bariatric surgery.

### Search strategy and selection criteria

A primary focus was on retrieving qualitative studies. However, qualitative parts of mixed method studies were also considered for inclusion [25]. A comprehensive database search was performed in February 2023. Articles published in English were identified from the following electronic databases: PubMed, CINAHL, Embase, PsycInfo, PsycArticles, Scopus, Cochrane Library and Web of Science. Furthermore, Open Grey was searched to find potential grey literature. The database searches were supplemented by hand-searching related reviews and the reference lists of the included studies.

Search terms (women, fertility, bariatric surgery, sexual function, obesity, sex hormones, experience, hermeneutics, qualitative research, vasomotor) and appropriate synonyms were applied in the electronic databases and available literature was browsed in order to become immersed in the available information. The search terms and queries were adjusted on a regular basis to ensure that the maximum possible number of reports could be acquired. Changes and modifications to the search terms and queries were registered. Reports that did not fulfil inclusion criteria were excluded. See Fig 1 for the Prisma flowchart concerning the screening process and S2 Table for the entire search history.

### Critical appraisal of full-text studies

The first step in the appraisal was to review the individual reports and check if they met the inclusion criteria. This helped us to understand the content and methodology of each report. After that, we compared the reports and created cross-study summaries to highlight the key elements of each report. We did this in preparation for integrating the report findings. Next,

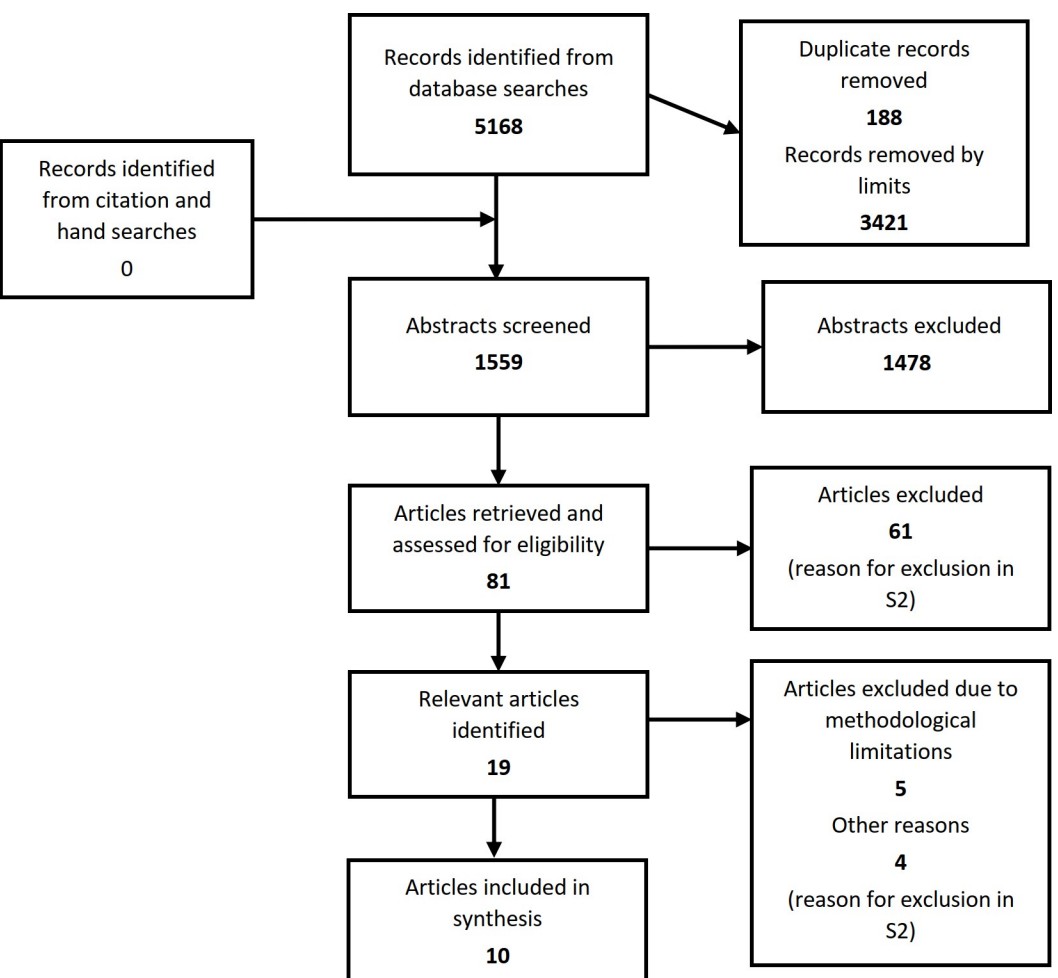

**Fig 1. Prisma flowchart reporting the process of identification, screening, and selection of papers for the meta-synthesis.**

we classified the findings of the qualitative research reports according to the typology of the reports. This helped us to determine which reports to include and exclude before performing a synthesis. We used the Primary Research Appraisal Tool-Qualitative (PRAT -Q) [26] to appraise the reports.

## Ethics statement

Application for ethical approval was not carried out as the study is a systematic review of already conducted and ethically approved studies. However, the study has been reviewed and registered with PROSPERO, registration number CRD42023394225 [21].

## Reflexive note

Maintaining reflexivity in qualitative analysis involves being aware of one's own preconceived notions and expectations. RP is a general surgeon who specializes in breast cancer surgery and is also a PhD student with a particular interest in women's health issues related to obesity, bariatric surgery, and sex hormone changes. CB is a qualitative researcher who focuses on cancer and chronic diseases, particularly in relation to women's health. CB is also experienced in

concept analyses and meta-synthesis. TO, EA and JF are bariatric surgeons with previous research experience in women's health and hormonal changes related to bariatric surgery. Applying a hermeneutical analysis approach benefits from the researchers' own experiences and understandings, which can be an advantage or even a prerequisite for the analysis [27].

## Data extraction and meta-synthesis

The primary researcher downloaded all of the reports to the EndNote X7, Thomson Reuters program. Two researchers (RP and CB) reviewed and discussed the reports, documenting all changes and decisions made. Two authors (RP and CB) independently extracted data from the included studies in duplicate, with continuous discussion and review. Data extraction was performed using a standardised data extraction form developed by CB. The data extraction form was piloted before use and used in other meta-synthesis studies [28–31]. Text related to the aim of this study was extracted, and the authors repeatedly discussed data extraction to ensure consensus and rigour. The extracted data varied as most papers presented raw data as direct quotations from participant interviews or interpreted text and reflections. Table 1 provides an example of the data extraction form. The four principles of Gadamer's hermeneutic analysis [27, 32] were performed from these statements from the studies' findings to evaluate the text, within the context of a guiding question: How does bariatric surgery affect premenopausal women's *womanliness* in terms of fertility and sexuality? 1) Data immersion was carried out by reading through the included studies repeatedly creating an understanding of the text as a whole. 2) An analysis of the text was carried out with meaningful excerpts of text from the study results being extracted and discussed (text horizon). 3) The interpreter's translation and understanding of the text excerpt was described and deliberated (interpreter horizon). 4) Horizons were merged into fusions presented in a whole text providing a broader and richer understanding of the phenomena exposed from the interpreted article data. The whole study proceeded as a hermeneutic circle, from the part to the whole and back again, study by study, sentence by sentence.

## Results

### Included studies

The initial search strategy resulted in5,168 articles that included different types of studies such as quantitative, qualitative, mixed-methods, case reports, consensus statements and abstracts.

**Table 1. Example of the data extraction form.**

| The horizon of the text | The horizon of the interpreter | The fusion | "Comment" |
|---|---|---|---|
| **1 Alleva et al 2022** | | | |
| Most women described the body as valuable in relation to important others, such as seeing one's children grow up. It is noteworthy that this theme was apparent in each of the three writing exercises, not just the writing exercise that focused on communication with others. | Maintaining a healthy body in order to live a long life | **A healthy and functioning body** | Be healthy- live long and having a family |
| **2 Condori et al 2019** | | | |
| Like, I've always had this ideal body, that I don't want to be super skinny, not at all, but a bit chubby, like, still having the curves. I don't like to be the way I am now, for example, that I'm like, overweight. . .. I've got an ideal body. It's just hiding, somewhere in here, right now. (Participant 3) | The obese body hinders the person to be the person it wants to be, having a body with curves, not skinny | **Womanliness** | Obesity is a hinder to be a woman in all senses |
| | | **A healthy and functioning body** | |
| **10 Young et al 2013** | | | |
| But you look at yourself after losing 20 pounds and you're like god I'm still fat, I still look at myself and see myself fat. And I can see the difference, I can see 20 pounds off of me but I still see a fat person. (14 days post-surgery) | Living with obesity affects your mental experience of body image | **Mind and body connection** | Adjusting the physical changes and mental adaptation |

After removing duplicate articles (188), 3,421 articles were filtered out based on specific inclusion criteria, such as studies exclusively featuring women participants, qualitative methods, and post-operative experiences, leaving only 1,559 articles. The abstracts of these 1,559 articles were assessed for relevance and 81 articles were further reviewed to ensure they met the inclusion/exclusion criteria. This led to 19 articles that met the necessary criteria, but four of them were excluded due to incomplete data. In the end, 15 articles were appraised according to PRAT-Q, and 10 studies published between 2010 and 2023 were included in the data analysis [33–42]. Generally, 10 to 20 studies are considered ideal for meta-synthesis [43]. The excluded studies are shown in S3 Table.

**Characteristics of included studies.** The ten studies included a total of 156 women. The studies used varying qualitative analysis methods such as thematic analysis [33–35, 41], discourse analysis [36, 42], phenomenology [37, 38] and qualitative content analysis [39, 40]. The studies were conducted in the Netherlands [33], Sweden [34, 35, 41], Italy [36], Turkey [37], Denmark [38], Brazil [39, 40] and New Zealand [42]. The study participants shared their experiences deriving from bariatric surgery and its effect on sex hormones, sexuality, and everyday life. See S4 Table concerning primary key features of the included studies.

## Confidence in synthesis findings

Once the fusions were identified and created the level of confidence in each fusion was assessed using the Grading of Recommendations Assessment, Development, and Evaluation–Confidence in Evidence from Reviews of Qualitative research (GRADE-CERQual) [44]. There are four domains in GRADE-CERQual that assess uncertainties in the data and include: methodological limitations, relevance, adequacy of data, and coherence. This assessment aims to evaluate and describe through a transparent procedure how much confidence there is in the findings. Therefore, an overall confidence rating of "high," "moderate," "low," or "very low" was assigned to each fusion, considering each of the four GRADE CERQual components.

All fusions were judged to have a high confidence level because they were based on substantial and robust material where the context was relevant, data was rich, had high coherence, and only minor methodological limitations. See Table 2 concerning findings and GRADE-CERQual assessment. In Table 2, we display the confidence of our study findings according to GRADE-CERQual on our results. This provides support that the results are grounded in the included studies, thereby representing a more significant number of women than the ordinary small sample size in qualitative studies.

## Meta-synthesis

The qualitative interpretative meta-synthesis resulted in three fusions: "Womanliness," A healthy and functioning body," and "Mind and Body Connection." The fusions are interdependent and interacting, thus bridging over to other fusions, therefore presented in the following order, reflecting a more profound and broadened understanding of women's experiences of bariatric surgery and its effects on their lives. See Fig 2 for an illustrative overview of the findings. In addition, the following presentation includes quotes from the studies that contributed to the creation of each fusion. See S1–S4 Datasets concerning the data analysis procedure and creation of fusions.

**Womanliness.** Womanliness is often described as having the traits, qualities and behaviors that a culture regards as especially characteristic of or ideally appropriate to adult women. Femininity can be associated with womanliness and is considered attributes and roles, often biologically and socially constructed, that are associated with women and girls. Women stated a return to what they perceived being femininity after the weight loss. The reduction of extra

**Table 2. Summary of the findings and their confidence.**

| Fusion | Content, articles contributing to the fusions and number of participants | Confidence in the findings according to CERQual |
|---|---|---|
| **Womanliness** | Womanliness is defined as having the physical characteristics and related attributes of an adult biological female. The reduction of extra weight with a return to female pattern fat distribution made participants feel womanlier. There was joy in purchasing clothing that brought attention to their new figure. This experience of womanliness motivated them to care for themselves health-wise and concerning their outer appearances. There was also an improvement in their relationships with their partners, who showed a new appreciation for them. Their increased self-esteem and positive mood motivated their partners to become closer to them. The women were no longer ashamed of their bodies and instead enjoyed sexual activity with their partners. Womanliness meant that there was a return to experiences of femininity when their hormones returned to physiological regulation and their fertility was restored, giving them the ability to become a mother. <br><br> [27–36] n = 156 | High confidence, there is a reasonable representation of the phenomenon under study. <br> Minor methodological limitations <br> High coherence, there is a clear fit between the data (10 studies) and the fusion. <br> Adequacy of data, there is rich and sufficient data to support the fusion. <br> Relevance, the context is consistent with the research question. |
| **A healthy and functioning body** | There was a restoration to a healthy body after surgery. The weight loss led to a decreased burden on their bodies, and they were able to exercise and activate themselves, an increase in energy and stamina allowed them to take part in life in a new way: both in psychical and social activities. A functioning body also did communicate to the women that there is a need to be more aware of the side effects of surgery and to be able to read the body's signals concerning meal sizes and proper food composition. The healthy and functioning body was appreciated and there was a clearer awareness of their body's physical signals and living a healthier lifestyle since they were more in tune with their physical state. <br><br> [27–29, 31–36] n = 126 | High confidence, there is a reasonable representation of the phenomenon under study. <br> Minor methodological limitations. <br> High coherence, there is a clear fit between the data (9 studies) and the fusion. <br> Adequacy of data, there is rich and sufficient data to support the fusion. <br> Relevance, the context is consistent with the research question. |
| **Mind and body connection** | This is the belief that our attitudes, thoughts and feelings have an impact on our body and health. This connection also works the other way around in that how we treat our body and physical attributes can influence our mental health. The women had often identified themselves as obese and had difficulty adapting themselves to a life as a normal-weight person There was a delay in understanding that they had lost weight and had a new appearance. The profound and often sudden weight loss created a severe change in their appearance that could challenge their opinion and acceptance of themselves. The women also experienced certain difficulties in dealing with how people and society treated them after weight loss. The women had previously belonged to groups that promoted pride in having a large body, now, they felt that they had let that group down and instead become part of the adversary. <br><br> [27–30, 32–36] n = 140 | High confidence, there is a reasonable representation of the phenomenon under study. <br> Minor methodological limitations <br> High coherence, there is a clear fit between the data (9 studies) and the fusion. <br> Adequacy of data, there is rich and sufficient data to support the fusion. <br> Relevance, the context is consistent with the research question. |

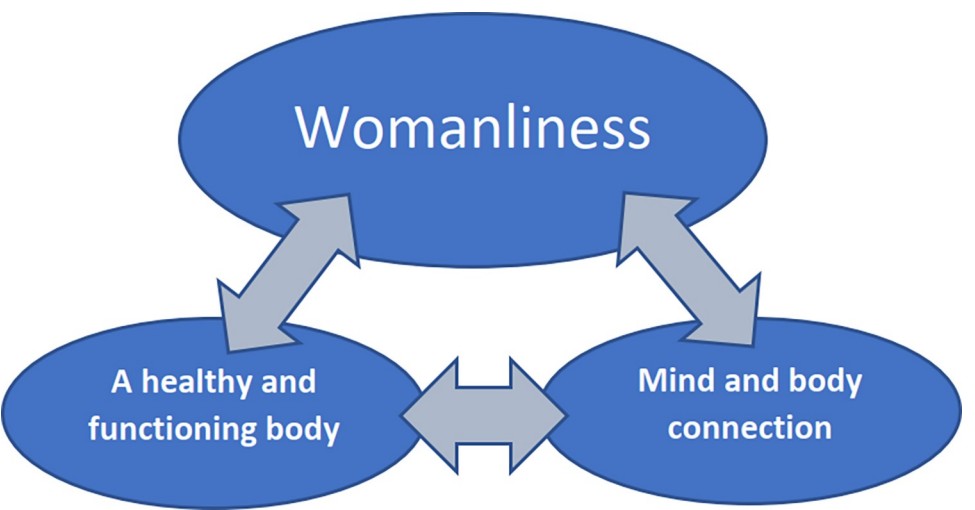

**Fig 2. Visualise the fusions.**

weight with a return to female pattern fat distribution made them feel womanlier and they enjoyed purchasing clothing that brought attention to their new figures. The experience of a return to womanliness made them more motivated to care for themselves health-wise and concerning their outer appearances. The weight loss made them strengthen their self-appreciation which lifted their self-confidence and motivated the women to express themselves, be assertive and even demand more of others. Before the surgery, the women hid within their excess weight and often portrayed a façade of joy and humour to hide their sadness. The weight loss was metaphorically likened to a cocoon being removed, and instead, they presented their true outer selves. A metamorphosis and new start in their lives occurred.

> *In contrast, almost a third of women described increased appearance evaluation and investment post-surgery, such as enjoying clothes shopping and feeling more satisfied with their appearance. Almost a third described social freedom and confidence and felt more accepted in social gatherings. [33]*

> *Women stated that weight loss and having a slim physical appearance gave them more freedom in what to wear and increased their self-esteem after surgery. Some women described the desire to look in the mirror and like what they see. [37]*

The increased self-esteem that the women experienced from their weight loss also improved their relationships. The women became more social and enjoyed contact with family, friends, and their partners in a completely different way. Prior to weight loss, the women often lacked the energy and endurance to take part in social activities. However, the changes after surgery and weight loss motivated the women and brought about positive emotions so that contact with other people was enjoyable. The women also felt an improvement in their relationships with their partners, who were positively influenced by the changes in their partner's moods and stamina. The women also felt that their partners showed a new appreciation for them and that their increased self-esteem and positive mood motivated their partners to become closer to them.

> *I want to date and be this new person. I think that's the struggle in maintaining a relationship, you feel like you got a second chance on life. You just reinvented yourself and you reinvented what you want, what you want out of life, what you expect from other people, what you expect from yourself. I think all of that stuff is different for me than a year ago. [42]*

> *We also found the positions: 'my family' 'my partner' and 'my work' were significant, while after the surgery, the relational component becomes even more important. [36]*

The women described a feeling of womanliness in wanting to take part in sexual activity, were no longer ashamed of their bodies, and instead enjoyed sexual activity with their partners which strengthened their relationships. They had previously avoided sexual activity, and shunned their partners, out of shame for the way that they looked and their physical attributes.

> *Most of the participants described a more active sex life, which was also more satisfying than before surgery. Internal factors, such as being more comfortable in a sexual situation and enhanced self-esteem, allowed them to demand more of their partners. [35]*

> *In addition to experiencing an increase in sexual desire after the surgery, some women stated that their husbands' sexual desire also increased and having more satisfaction due to having more options in movements and motion during intercourse. [37]*

The women felt a return to normality and womanliness which likely was related to restored sex hormone levels. The weight loss and normalization of sex hormones became obvious to the women as their menstrual cycles became regular and they experienced ovulation symptoms. The women now had hope that they would be able to achieve their dreams of becoming pregnant and creating a family having their fertility restored. They felt that undergoing surgery was an important step and solution to their previous difficulties in becoming pregnant. They also described the satisfaction of finally feeling like women when they had regular menstruation, felt their ovulations, and had improved sexual function.

*Another participant pointed out less need for lubricants as the cycle had become more regular and ovulatory. Yet another said that the weight loss had increased "the feeling of having intercourse." [35]*

*Before I felt that I had a lack of female hormones, I felt that nothing worked as it should, and now, when everything works, it feels like a whole new world, I feel like a woman now. [41]*

*Through bodily, social, and mental well-being, all participants considered the opportunity of realizing their dreams of having children and raising a family. [38]*

There were some negative experiences related to weight loss and relationships. The women experienced that their husbands could express jealousy and fear that other people would find their wives attractive. There became an imbalance in the relationship when the women became empowered with self-esteem and confidence after their weight loss. Certain partners expressed a fear that their wives would leave them after losing weight and gaining self-assurance. Some women had the perception that the weight loss's positive qualities could lead to more destructive experiences in their relations.

*After all the effort to find their place in the world again and to be admired, they have faced the reappearance of their feminine bodies: a new situation with which they are unable to cope. [40]*

*In the postoperative period, the women have to face new life experiences, such as jealousy, mistrust, fear, and envy that, until recently, had not existed. [39]*

**A healthy and functioning body.** The women experienced a restoration to a healthy body after surgery. The weight loss led to a decreased burden on their bodies, and they were able to exercise and activate themselves. The women experienced an increase in energy and stamina that allowed them to take part in life in a new way. They were able to be active parents and enjoyed their children. Feelings of guilt no longer impeded them for not being able to be involved in their family lives. Those not yet having children now felt prepared to take on the role of an engaged mother.

*Participant 45: "Before the operation, I could not experience much fun. I was always tired and not in the mood to go somewhere. Now I have lost 27 kilos. I enjoy going out with my husband and daughter, especially because I am less tired." [33]*

The weight loss and newfound appreciation of their bodies made them more energized to take part in social activities. Being more social and open to others reduced feelings of isolation and abandonment that they had previously experienced prior to surgery. The weight loss

made it possible for the women to express themselves in a way that they had previously not had the courage to do.

*"Now, I no longer feel like it's uncomfortable to go exercising among other people, like they'd be thinking "what is she doing" and so on. I can do that. I am much, much more comfortable in social contexts, like I said. Going to birthday parties and, like hanging out with friends and so on, that feels great too." [35]*

*The first source of relief, after postsurgical recovery, comes from a strong sensation of acceptance and social reinsertion. They feel that they are part of a world which they were not a part of. They experience a feeling of genuine happiness. [39]*

The women realised a need to be more aware of the side effects of surgery and to be able to read the body's signals concerning meal sizes and proper food composition. The women were also concerned with the risk of regaining weight after surgery and experienced a fear of losing the joy and relief that they experienced after weight loss. There was a need to maintain control of their situation and not let their past return to them.

*P1: I hope that I manage to stay on the right path, and I believe in that. When you have become so pleased with yourself and you know how it would be if you fell back, then. . .. [38]*

The change in their bodies made it possible to see life and live in a positive way. They appreciated their bodies and became more aware of their body functions, describing a clearer awareness of their body's physical signals and living a healthier lifestyle since they were more in tune with their physical state.

*I have control over what goes in my mouth and what happens with my body now and that's the best thing that's happened from this surgery besides the weight loss. [42]*

**Mind and body connection.**   The women felt that they often had identified themselves as obese and had difficulty adapting themselves to a life as a normal-weight person. The psychological effects of living with obesity could remain and continue to affect their mood and opinions of themselves. Many described a need for psychological support to deal with the changes, often long after the surgery. They could often feel discomfort in public, fearing that others may scrutinize them even though they had attained a normal weight.

*Over a third of participants expressed appearance concerns, such as worries about excess skin and hair loss. A few women described that they had not adjusted to their slimmer body and expected to see a heavier person in the mirror. Others explained that they found it difficult to care for their appearance after years of self-loathing. [33]*

*You're actually not hungry when you eat. Your brain keeps telling you that you are hungry. The stomach on the contrary is about to burst. . .and it's hard to get rid of because your brain was not operated on. This need, it's not just removed in surgery. . ...There's such a psychological need, all the time. [38]*

The women experienced a delay in understanding that they had lost weight and had a new appearance. When they finally understood the change, they took pride in shopping and presenting their bodies with new clothes. They also saw their changed appearance as a new

beginning and a chance to make a new start in their lives and health. They used their new identity to create a newer, healthier, and better-functioning version of themselves.

*Several of the participants said that they no longer had thoughts of what other people might think of them and their body. This had previously restricted them in several everyday areas of their lives. Reflections about how it might have been all in their own head before came up, but nonetheless they now felt liberated from these worries. [35]*

*'I lived with difficulty', 'I remember moments of collapse', 'I threw my life to the wind', 'I tried, and it did not change anything'. The experiences related to the preoperative period are, therefore, very similar in the two groups; however, one year after, the person describes herself differently. A new position related to the action emerged to the perception of ability and to the will of dealing with new situations. Participants now stated: 'I assert myself' or 'I explore myself,' 'I can finally get out', 'I can go shopping,' 'I can be myself'. [36]*

*P3: I'm actually not the same person at all, as I was before surgery. Now, it's me that means something, this means that I will continue being the person I am now. [38]*

The women also experienced certain difficulties in dealing with how people and society treated them after weight loss. People could express jealousy at their ability to lose weight or envy their new figures. They also felt despair in experiencing the changed attitudes of others towards them after they lost weight and seeing clearly how differently people are treated based on their body size and appearance.

*"Before they did not like me because I was fat. Today I am thin . . .they will think that I am stealing their scene! I did not operate for this . . ." [39]*

*I had to adapt to my new shell even though I was the same person on the inside, the people around me, their focus changed towards me, their way of talking to me, socialising with me, it was a bit tough to get used to. [41]*

The sudden weight loss and subsequent change in appearance could have profound effects. The women dealt not only with their own identities but also with belonging to a certain group. The women had identified themselves as obese for a large portion of their lives and accepted certain factors related to living with obesity. They often lived with shame, fear and depression related to the struggles of living with reduced physical capacity, health problems and the stigma from society. The profound and often sudden weight loss created a severe change in their appearance that could challenge their opinion and acceptance of themselves.

*I am an ex-obese (. . .) When I look in the mirror, I see myself and I feel good-looking,. . ., when I try a dress, I can feel good. . . But when I think to myself, I guess I'm still chubby. . . maybe it happens to me to see some girls around and to think. . ., such a beautiful body, that envy! [36]*

The women had previously belonged to groups that promoted pride in having a large body and admonishing a need to fit in with a thin body. They felt that they had let that group down and instead become part of the adversary, at the same time that they enjoyed their weight loss and new figures. They needed to find a way to accept their new group and identity without disappointment.

*Like when I see fat girls, I'm like what's up 'cause you're my people. Like there's this really fat girl in my maths class and I just want to hang out with her and talk like fat girl stuff! But then*

*I feel like a fucking traitor because I was able to have surgery and lose a bunch of weight and now, I'm not like obese anymore. I just feel like I betrayed them kind of. [42]*

## Discussion

This systematic review and qualitative meta-synthesis aimed to improve the understanding of the life changes that occur in women who have undergone bariatric surgery, focusing on experiences of womanliness. Overall, our interpretation of 10 articles provided greater insight into the meaning of womanliness, aspects of a functioning body, and the mind and body connection after weight loss following bariatric surgery.

The women in the included articles experienced positive effects of restored womanliness related to normalised sex hormones, fertility and sexual function which are consistent with results of previous quantitative meta-analyses and reviews [45, 46]. Women also experienced improved sexual function which resulted in beneficial effects on quality of life [47] after weight loss from bariatric surgery. The importance of a functioning body with improved mobility after surgery and weight loss, as well as control of weight has previously been described in quantitative and qualitative studies [48, 49]. Previous quantitative studies also explored the women's descriptions of improved body image and beneficial effects on self-esteem and psychological well-being after surgery and subsequent weight loss [50, 51].

Findings concerning difficulties after surgery and weight loss relating to adapting to a new identity, dealing with fat bias and needs for post-operative support have also previously emerged from qualitative investigations [52–54] as well as in quantitative reviews [55, 56] and illustrate the need for routine preoperative information and selection as well as structured follow-up to support patients´ adaptation to life changes after surgery.

Participants in the analysed study reported that womanliness represents a return to a functioning body that enables an opportunity to become mothers, actively take part in their children's lives, and rediscover sexuality and intimacy with their partners. They report that these changes associated with restored womanliness improved women's quality of life as they could achieve their goals and strengthen relationships around them.

### Strengths and limitations

This systematic review and qualitative interpretative meta-synthesis utilised rigorous and standardised methods for appraisal, and inclusion is supported by a systematic approach. The eligible articles were reviewed using PRAT-Q [18], which entailed the design, selection, data collection, analysis, ethics, and theoretical frame of reference. Qualitative studies of women with obesity and particularly postoperative experiences after bariatric surgery is a sparsely investigated area, however, through careful appraisal, ten articles were found that could be included in this meta-synthesis and this strengthened the variety of material that was included in the interpretation. The studies represented women from various countries and cultural backgrounds providing a richer and more transferable description of this population's experience. Limitations are the possibility that articles may have been excluded based on limitations in the search terms, however repeated attempts to locate citations by hand did not increase the number of new articles received. There is a risk that language restraints excluded articles that may have provided new and more in-depth detail to our meta-synthesis. Another limitation is the generalisability of the findings to persons that do not identify with the aspects of womanliness and femininity that have been presented in this study. Also, post-operative experiences after weight loss were the focus of the study, and no consideration was taken to specific time intervals after surgery.

## Implications

Women with obesity contemplating undergoing bariatric surgery can be supported by the results in this meta-synthesis. Participants in the included studies reported mainly positive effects following surgery including a restored womanliness that includes improved fertility, sexual and physical function, and quality of life. However, we also report challenges for women after undergoing surgery and weight loss including aspects of a new "identity." We believe that focused support may be helpful for women in relation to undergoing bariatric surgery to ease the adjustment to physical and mental changes. Future prospective studies, preferably randomised, may evaluate if improved preoperative information concerning expectations and changes after surgery and an augmented support during follow-up could improve health-related quality of life in women undergoing bariatric surgery.

## Supporting information

**S1 Table. Prisma checklist.**
(DOCX)

**S2 Table. Search history.**
(DOCX)

**S3 Table. Excluded studies.**
(DOCX)

**S4 Table. Primary key features of the included studies.**
(DOCX)

**S1 Dataset. Analysis step 1.**
(DOCX)

**S2 Dataset. Analysis step 2.**
(DOCX)

**S3 Dataset. Analysis step 3.**
(DOCX)

**S4 Dataset. Analysis step 4.**
(DOCX)

## Acknowledgments

We would like to extend our gratitude to Linköping University Library librarians Isolina Ek and Katarina Jonzon for guidance and assistance during the literature search.

## Author Contributions

**Conceptualization:** Rebecca Paul, Ellen Andersson, Torsten Olbers, Jessica Frisk, Carina Berterö.

**Data curation:** Rebecca Paul, Carina Berterö.

**Formal analysis:** Rebecca Paul, Carina Berterö.

**Funding acquisition:** Rebecca Paul.

**Investigation:** Rebecca Paul, Carina Berterö.

**Methodology:** Rebecca Paul, Ellen Andersson, Torsten Olbers, Jessica Frisk, Carina Berterö.

**Project administration:** Rebecca Paul, Ellen Andersson, Torsten Olbers, Jessica Frisk, Carina Berterö.

**Resources:** Rebecca Paul, Torsten Olbers.

**Software:** Rebecca Paul.

**Supervision:** Carina Berterö.

**Validation:** Rebecca Paul, Carina Berterö.

**Visualization:** Rebecca Paul, Carina Berterö.

**Writing – original draft:** Rebecca Paul, Carina Berterö.

**Writing – review & editing:** Rebecca Paul, Ellen Andersson, Torsten Olbers, Jessica Frisk, Carina Berterö.

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
