## [Decision Letter · Decision Letter 0]

20 Dec 2023

PONE-D-23-14837Impact of bariatric surgery on premenopausal women’s womanliness: A qualitative systematic review and meta-synthesisPLOS ONE

Dear Dr. Paul,

Thank you for submitting your manuscript to PLOS ONE. After careful consideration, we feel that it has merit but does not fully meet PLOS ONE’s publication criteria as it currently stands. Therefore, we invite you to submit a revised version of the manuscript that addresses the points raised during the review process.

The manuscript has been evaluated by two reviewers, and their comments are available below.

The reviewers have raised a number of major concerns. They request improvements to the reporting of methodological aspects of the study, for example, regarding the exclusion criteria and more information on how the data collection was completed. The reviewers also note concerns about the small sample sizes in the included studies.

Could you please carefully revise the manuscript to address all comments raised?

We look forward to receiving your revised manuscript.

Kind regards,

Jennifer Tucker, PhD

Associate Editor

PLOS ONE

“I have read the journal's policy and the authors of this manuscript have the following competing interests: TO participated in advisory boards and educational activities for Johnson & Johnson and Novo Nordisk unrelated to the submitted article, and reimbursements were directed to his academic institution. RP, EA, JF and CB declare that no competing interests exist.”

Reviewers' comments:

Reviewer's Responses to Questions

**Comments to the Author**

1. Is the manuscript technically sound, and do the data support the conclusions?

Reviewer #1: Partly

Reviewer #2: Yes

2. Has the statistical analysis been performed appropriately and rigorously? 

Reviewer #1: N/A

Reviewer #2: N/A

3. Have the authors made all data underlying the findings in their manuscript fully available?

Reviewer #1: Yes

Reviewer #2: Yes

4. Is the manuscript presented in an intelligible fashion and written in standard English?

Reviewer #1: Yes

Reviewer #2: Yes

5. Review Comments to the Author

Reviewer #1: Dear authors,

Thanks for the submission of a manuscript describing an important and rarely discussed issue about bariatric surgery.

Although it is an important issue, I, as a bariatric surgeon, had a hard time to read your manuscript because it is too long and it is not the type of manuscript we are used to.

I understand that this is a qualitative research, but I think that you to make it shorter to make it easier to read and understand.

Why did you choose "particularly bypass"? I think that for the scope of your work other types of surgery could be included, as Sleeve for example.

What did you mean by "through limits another 3,421 articles were removed"?

Although the science behind meta-synthesis says: "Approximately 10 to 20 studies represent an ideal meta-synthesis", I am concerned about your data using only 10 papers from 5,168 and, many of them with very few patients. Can you better explain and justify it beyond GRADE-CERQual?

Table 1 is very busy and makes your manuscript longer. Is it really necessary?

Table 2 should be transformed in text. As a table, it is hard to read.

Reviewer #2: This manuscript does an excellent job in summarizing and synthesizing existing qualitative studies on women undergoing bariatric surgery and their experiences of weight loss and womanliness. The findings are clearly presented and make sense when reading the included studies. The meta-synthesis contributes to the understanding of benefits to women’s health and well-being, as well as the difficulties in adapting mentally to changes in outer appearance after bariatric surgery.

Major issues

Methods

162 I'd like to know a bit more about how the data extraction was made. Did you not use any software? It might be clarifying to see an example of the data extraction form as an illustration.

Table 1

I do not understand why the references have different numbering here and in the reference list. This makes it difficult to follow as to which study supports which findings.

Minor issues

Scrutinize language!

Line

61-62 This sentence is difficult to understand. 63 having obesity? 116 utilising? 426 This sentence is difficult to understand.

433 “a” word missing? 561 Reference needs check-up.

6. PLOS authors have the option to publish the peer review history of their article (what does this mean?). If published, this will include your full peer review and any attached files.

Reviewer #1: **Yes: **CARLOS AURELIO SCHIAVON

Reviewer #2: No

---

## [Author Response · Author response to Decision Letter 0]

15 Jan 2024

Jennifer Tucker, PhD

Associate Editor

PLOS ONE

Dear Dr. Tucker,

Thank you for the valuable feedback and comments. We have made several changes according to reviewers’ suggestions and feel that our manuscript has been greatly improved. 

1. Thank you for highlighting necessary adjustments concerning style requirements, we have reviewed our manuscript and corrected any deviations from the described instructions.

2. We include the following statement to our Competing Interests section since we have no restrictions on sharing of data and materials: “This does not alter our adherence to PLOS ONE policies on sharing data and materials.” 

Reviewers’ comments 

1. Thank you for your responses.

2. Thank you for your responses.

3. Thank you for your responses.

4. Thank you for your responses.

5. Reviewer #1

Why did you choose "particularly bypass"? I think that for the scope of your work other types of surgery could be included, as Sleeve for example.

Thank you for your relevant remark. We chose a primary focus on gastric bypass since this procedure has, until relatively recently, been the most common procedure in bariatric surgery and our previous studies included patients having undergone Roux-en-Y gastric bypass surgery. 

We did, however, include studies that involved other forms of bariatric surgery such as vertical sleeve gastrectomy and gastric banding in order to broaden our findings. As sleeve gastrectomy is the most prevalent bariatric procedure worldwide, we do agree that it would have been valuable to include more studies with patients who have undergone this procedure. However, the only relevant qualitative studies found in this review predominantly involved gastric bypass surgery with a minority of participants having undergone sleeve or banding. This may have been a consequence of our inclusion criteria involving qualitative research method, exclusively female participants and with focus on postoperative experiences.

What did you mean by "through limits another 3,421 articles were removed"?

We applied limits based on our inclusion criteria such as “women” “humans” and publishing dates that reduced the included total of articles by 3,421. The inclusion criteria of exclusively premenopausal women, publication date 2000-2023 and exclusively qualitative or mixed methods studies also greatly reduced the number of relevant studies that could be included in the analysis.

Although the science behind meta-synthesis says: "Approximately 10 to 20 studies represent an ideal meta-synthesis", I am concerned about your data using only 10 papers from 5,168 and, many of them with very few patients. Can you better explain and justify it beyond GRADE-CERQual?

We fully understand the reviewer’s concern. However, the search strategy was broad in order to not miss any relevant papers. The few studies involving qualitative research method, exclusively female participants and focus on postoperative experiences led to the reduction from 5,168 studies to 10 relevant articles that correspond to our inclusion criteria. The ten included studies utilizing findings accumulated from 156 women. The data included in the meta-synthesis are quotations directly from participants and the researcher’s interpretation. Text excerpts were extracted and reviewed prior to meta-synthesis in order to determine if a necessary richness and depth of data was available in order to carry out a meta-synthesis. We determined that the amount of data from the 10 included studies was sufficiently abundant with data richness to allow an analysis that involved the four criteria of trustworthiness in qualitative research (dependability, credibility, confirmability, and transferability [1]. In Table 2, we display the confidence of our study findings according to GRADE-CERQual for our results. This provides evidence that the result is grounded in the included studies, thereby representing a more significant number of women than the ordinary small sample size in qualitative studies. 

Table 1 is very busy and makes your manuscript longer. Is it really necessary?

We appreciate your comments. Table 1 has been removed from the manuscript and included as a supplement (S4) in order to shorten the manuscript and make it more concise.

Table 2 should be transformed in text. As a table, it is hard to read.

Thank you for your comments, and we appreciate the large amount of text involved in the descriptions in Table 2. However, we feel it is necessary to include Table 2 in the manuscript in order to provide easily accessible transparency concerning our findings.

Reviewer #2

Major issues

Methods

162 I'd like to know a bit more about how the data extraction was made. Did you not use any software? It might be clarifying to see an example of the data extraction form as an illustration.

Thank you for your question concerning data extraction. We did not use computer software for the data extraction process. We utilized a standardized data extraction form that has been used in previous studies and references to these studies have been provided. We have included an example (Table 1) in order to facilitate understanding of the data extraction process.

Table 1

I do not understand why the references have different numbering here and in the reference list. This makes it difficult to follow as to which study supports which findings.

Thank you for noting this error which occurred during formatting. We have now made the necessary adjustments and revised Table 1 as supplementary information S4. Since Table 1 has become supplementary information S4 auxiliary to the manuscript, the changes are not marked with the tracking changes tool.

Minor issues

Scrutinize language! 

Line

61-62 This sentence is difficult to understand. 63 having obesity? 116 utilising? 426 This sentence is difficult to understand.

433 “a” word missing? 561 Reference needs check-up.

Thank you for your comments, we have scrutinized the language and made several modifications as well as the suggested adjustments and clarifications. The reference has been amended.

On behalf of all authors,

Rebecca Paul, M.D., Ph.D.

Department of Biomedical and Clinical Sciences, Linköping University, Sweden

 

References

1. Guba EG. Criteria for assessing the trustworthiness of naturalistic inquiries. ECTJ. 1981;29(2):75. doi: 10.1007/BF02766777.

---

## [Decision Letter · Decision Letter 1]

27 Feb 2024

PONE-D-23-14837R1Impact of bariatric surgery on premenopausal women’s womanliness: A qualitative systematic review and meta-synthesisPLOS ONE

Dear Dr. Paul,

Thank you for submitting your manuscript to PLOS ONE. After careful consideration, we feel that it has merit but does not fully meet PLOS ONE’s publication criteria as it currently stands. Therefore, we invite you to submit a revised version of the manuscript that addresses the points raised during the review process. 

We look forward to receiving your revised manuscript.

Kind regards,

Annesha Sil, PhD

Associate Editor, PLOS ONE

Journal Requirements:

**Additional Editor Comments:** After careful reviewing, before we can recommend acceptance, we would kindly request you to address a minor outstanding concern regarding terminology. Specifically, we note that the term "womanliness" appears throughout the manuscript. However, we note that the term does not appear to be associated with a concrete definition, and moreover, several themes as discussed in the womanliness section (beginning ln 228) do not appear to match the provided definition.

We would therefore kindly request that you update your submission to either a) replace or qualify the theme 'womanliness' (as stated on ln 228) with a description that more broadly encompasses the themes discussed in this section, or b) create an additional section that encompasses the themes, in addition to the existing header 'womanliness' that is based on the definition provided and labelled accordingly. Please do not hesitate to reach out to us if you have any further concerns or queries regarding our revision request, and we look forward to hearing from you.

Reviewers' comments:

Reviewer's Responses to Questions

**Comments to the Author**

1. If the authors have adequately addressed your comments raised in a previous round of review and you feel that this manuscript is now acceptable for publication, you may indicate that here to bypass the “Comments to the Author” section, enter your conflict of interest statement in the “Confidential to Editor” section, and submit your "Accept" recommendation.

Reviewer #1: All comments have been addressed

2. Is the manuscript technically sound, and do the data support the conclusions?

Reviewer #1: Yes

3. Has the statistical analysis been performed appropriately and rigorously? 

Reviewer #1: I Don't Know

4. Have the authors made all data underlying the findings in their manuscript fully available?

Reviewer #1: Yes

5. Is the manuscript presented in an intelligible fashion and written in standard English?

Reviewer #1: Yes

6. Review Comments to the Author

Reviewer #1: I have no conflict of interest.

Dear authors,

Thank s for your responses. The manuscript improved a lot.

7. PLOS authors have the option to publish the peer review history of their article (what does this mean?). If published, this will include your full peer review and any attached files.

Reviewer #1: **Yes: **Carlos Aurelio Schiavon

---

## [Author Response · Author response to Decision Letter 1]

17 Apr 2024

Annesha Sil, PhD

Associate Editor, PLOS ONE

Dear Dr. Sil,

Thank you for your careful review and recommendations concerning our manuscript, 

 " Impact of bariatric surgery on premenopausal women's womanliness: A qualitative systematic review and meta-synthesis." Your recommendations, previous editors, and reviewers have dramatically improved our paper. Below, we respond to comments.

Journal Requirements:

 Please review your reference list to ensure that it is complete and correct. If you have cited papers that have been retracted, please include the rationale for doing so in the manuscript text or remove these references and replace them with relevant current references. Any changes to the reference list should be mentioned in the rebuttal letter that accompanies your revised manuscript. If you need to cite a retracted article, indicate the article's retracted status in the References list and also include a citation and full reference for the retraction notice.

We have reviewed our references and found none that have been retracted. We did find an erratum in one reference, which we have included in the reference list.

 Additional Editor Comments:

After careful reviewing, before we can recommend acceptance, we would kindly request you to address a minor outstanding concern regarding terminology.

Specifically, we note that the term "womanliness" appears throughout the manuscript. However, we note that the term does not appear to be associated with a concrete definition, and moreover, several themes as discussed in the womanliness section (beginning ln 228) do not appear to match the provided definition.

 We would therefore kindly request that you update your submission to either a) replace or qualify the theme 'womanliness' (as stated on ln 228) with a description that more broadly encompasses the themes discussed in this section, or b) create an additional section that encompasses the themes, in addition to the existing header 'womanliness' that is based on the definition provided and labelled accordingly.

We have now clarified and defined the term womanliness in order to enable a clearer understanding of the fusion resulting from the meta-analysis of the included studies.

 Reviewers' comments:

Comments to the Author 1-7

Thank you for reviewing our manuscript and providing useful recommendations that have greatly improved it.

On behalf of all authors,

Rebecca Paul, M.D., Ph.D. Department of Biomedical and Clinical Sciences, Linköping University, Sweden

---

## [Editor Report · Decision Letter 2]

14 May 2024

PONE-D-23-14837R2Impact of bariatric surgery on premenopausal women’s womanliness: A qualitative systematic review and meta-synthesisPLOS ONE

Dear Dr. Paul, Thank you for submitting your revised manuscript with an updated definition regarding the terminology ‘womanliness’. Before we can recommend acceptance, the editorial team still has a few outstanding concerns that we require you to address. Therefore, we invite you to submit a revised version of the manuscript that addresses these points. 

The editorial team still has some concerns regarding the definition of ‘womanliness’ as provided in ln 230-233 as this is not entirely clear and several themes in the manuscript do not appear to match the provided definition and can be seen to seem exclusionary for some groups, which limits the generalizability of your conclusions.

We would therefore ask you to update the limitations section and Table 2 to outline how the definition of womanliness might have varied between the various participants and studies analyzed in this systematic review. 

Further, as this is a qualitative study that primarily analyses the reported experiences of women in response to bariatric surgery, we are concerned that some of the conclusions made in the study might be overstated or overinterpreted. Accordingly, we kindly request you to revise these as per our suggestions below:

Line 51: ‘Bariatric surgery has several benefits to women’s health and well- being, although difficulties in adapting mentally to changes in outer appearance need to be managed in order to successfully move forward with a new womanly life after surgery.

We suggest that you update this as follows:  "Women that have undergone bariatric surgery report several benefits to their health and well-being…". Additionally, we suggest that you remove the word ‘womanly’ at the end of the sentence (new ‘womanly’ life) or revise it to clarify exactly what is meant by this. 

Line 433: ‘Womanliness represents a return to a functioning body that enables an opportunity to become mothers, actively take part in their children’s lives, and rediscover sexuality and intimacy with their partners. These changes associated with restored womanliness improved women’s quality of life as they could achieve their goals and strengthen relationships around them.’

We suggest that you update this as follows: ‘Participants in the analysed study report that womanliness represents a return to a functioning body that enables an opportunity to become mothers, actively take part in their children’s lives, and rediscover sexuality and intimacy with their partners. They report that these changes associated with restored womanliness improved women’s quality of life as they could achieve their goals and strengthen relationships around them.’ 

Line 454: ‘Women with obesity contemplating undergoing bariatric surgery can be supported by the results in this meta-synthesis showing mainly positive effects following surgery including a restored womanliness that includes improved fertility, sexual and physical function, and quality of life.’

We suggest you update this line as follows: ‘Women with obesity contemplating undergoing bariatric surgery can be supported by the results in this meta-synthesis; participants in the included studies reported mainly positive effects following surgery including a restored womanliness that includes improved fertility, sexual and physical function, and quality of life.’

We appreciate your attention to these requests. Please do not hesitate to reach out to us if you have any further concerns or queries regarding our revision requests, and we look forward to hearing from you.

We look forward to receiving your revised manuscript.

Kind regards,

Annesha Sil, Ph.D.

Associate Editor

PLOS ONE
---

## [Author Response · Author response to Decision Letter 2]

9 Jun 2024

Annesha Sil, PhD

Associate Editor, PLOS ONE

Dear Dr. Sil,

Thank you for your careful review and recommendations concerning our manuscript, PONE-D-23-14837R2, "Impact of bariatric surgery on premenopausal women's womanliness: A qualitative systematic review and meta-synthesis." Your current and previous recommendations, as well as those from previous editors and reviewers, have dramatically improved our paper. The following are our responses to your feedback and recommendations.

Editor Comments:

The editorial team still has some concerns regarding the definition of ‘womanliness’ as provided in ln 230-233 as this is not entirely clear and several themes in the manuscript do not appear to match the provided definition and can be seen to seem exclusionary for some groups, which limits the generalizability of your conclusions.

We would therefore ask you to update the limitations section and Table 2 to outline how the definition of womanliness might have varied between the various participants and studies analyzed in this systematic review..

We agree with your concerns and also feel that it is important to clarify the generalizability of the findings in our study. We have now adjusted the limitations section as well as Table 2 in order to focus on the relevant population.

Editor suggestions line 51:

Line 51: ‘Bariatric surgery has several benefits to women’s health and well- being, although difficulties in adapting mentally to changes in outer appearance need to be managed in order to successfully move forward with a new womanly life after surgery.

We suggest that you update this as follows: "Women that have undergone bariatric surgery report several benefits to their health and well-being…". Additionally, we suggest that you remove the word ‘womanly’ at the end of the sentence (new ‘womanly’ life) or revise it to clarify exactly what is meant by this. 

We appreciate your suggestion and feel that it improves and clarifies the intent of our conclusion. We have made the necessary alterations in the abstract.

Editor suggestions line 433:

Line 433: ‘Womanliness represents a return to a functioning body that enables an opportunity to become mothers, actively take part in their children’s lives, and rediscover sexuality and intimacy with their partners. These changes associated with restored womanliness improved women’s quality of life as they could achieve their goals and strengthen relationships around them.’

We suggest that you update this as follows: ‘Participants in the analysed study report that womanliness represents a return to a functioning body that enables an opportunity to become mothers, actively take part in their children’s lives, and rediscover sexuality and intimacy with their partners. They report that these changes associated with restored womanliness improved women’s quality of life as they could achieve their goals and strengthen relationships around them.’ 

We agree with the proposed text and feel that it clarifies the meaning and experiences of womanliness. The necessary changes have been made to the text.

Editor suggestions line 454:

Line 454: ‘Women with obesity contemplating undergoing bariatric surgery can be supported by the results in this meta-synthesis showing mainly positive effects following surgery including a restored womanliness that includes improved fertility, sexual and physical function, and quality of life.’

We suggest you update this line as follows: ‘Women with obesity contemplating undergoing bariatric surgery can be supported by the results in this meta-synthesis; participants in the included studies reported mainly positive effects following surgery including a restored womanliness that includes improved fertility, sexual and physical function, and quality of life.’ 

Thank you for your recommendation concerning alteration of the first line in the Implications section, we have made the adjustment with the provided text.

On behalf of all authors,

Rebecca Paul, M.D., Ph.D. Department of Biomedical and Clinical Sciences, Linköping University, Sweden

---

## [Editor Report · Decision Letter 3]

17 Jul 2024

Impact of bariatric surgery on premenopausal women’s womanliness: A qualitative systematic review and meta-synthesis

PONE-D-23-14837R3

Dear Dr. Paul,

We’re pleased to inform you that your manuscript has been judged scientifically suitable for publication and will be formally accepted for publication once it meets all outstanding technical requirements.

Kind regards,

Laura Kelly

Division Editor

PLOS ONE
---

## [Editor Report · Acceptance letter]

18 Jul 2024

PONE-D-23-14837R3 

PLOS ONE

Dear Dr. Paul, 

I'm pleased to inform you that your manuscript has been deemed suitable for publication in PLOS ONE. Congratulations! Your manuscript is now being handed over to our production team.

Kind regards, 

on behalf of

Dr. Laura Hannah Kelly 

Staff Editor

PLOS ONE